# Scientific Rationale for the Treatment of Cognitive Deficits from Long COVID

Arman Fesharaki Zadeh [1,2,*], Amy F. T. Arnsten [3] and Min Wang [3,*]

1     Departments of Neurology, Yale University School of Medicine, New Haven, CT 06510, USA
2     Departments of Psychiatry, Yale University School of Medicine, New Haven, CT 06510, USA
3     Departments of Neuroscience, Yale University School of Medicine, New Haven, CT 06510, USA;
      amy.arnsten@yale.edu
*     Correspondence: arman.fesharaki@yale.edu (A.F.Z.); min.wang@yale.edu (M.W.)

**Abstract:** Sustained cognitive deficits are a common and debilitating feature of "long COVID", but currently there are no FDA-approved treatments. The cognitive functions of the dorsolateral prefrontal cortex (dlPFC) are the most consistently afflicted by long COVID, including deficits in working memory, motivation, and executive functioning. COVID-19 infection greatly increases kynurenic acid (KYNA) and glutamate carboxypeptidase II (GCPII) in brain, both of which can be particularly deleterious to PFC function. KYNA blocks both NMDA and nicotinic-alpha-7 receptors, the two receptors required for dlPFC neurotransmission, and GCPII reduces mGluR3 regulation of cAMP-calcium-potassium channel signaling, which weakens dlPFC network connectivity and reduces dlPFC neuronal firing. Two agents approved for other indications may be helpful in restoring dlPFC physiology: the antioxidant N-acetyl cysteine inhibits the production of KYNA, and the $\alpha$2A-adrenoceptor agonist guanfacine regulates cAMP-calcium-potassium channel signaling in dlPFC and is also anti-inflammatory. Thus, these agents may be helpful in treating the cognitive symptoms of long COVID.

**Keywords:** COVID-19; cognitive; brain fog; prefrontal cortex; n-acetyl cysteine; guanfacine

## 1. Introduction

Many patients are experiencing sustained cognitive deficits as a consequence of COVID-19 infection, a phenomenon often referred to colloquially as "brain fog", as part of long COVID. The World Health Organization states that "approximately 10–20% of people experience a variety of mid- and long-term effects after they recover from their initial illness. These mid- and long-term effects are collectively known as post-COVID-19 condition or "long COVID." Accumulating data indicate that the cognitive functions of the prefrontal cortex (PFC), including working memory and the executive functions, are the most consistently impaired in long COVID. These cognitive deficits can be sufficiently debilitating to interfere with the ability to work and/or care for families. Although there is an urgent need for treatment, there are currently no FDA-approved treatments for long COVID cognitive deficits.

The current review describes strategies for pharmacological treatment of long COVID cognitive deficits using medications already approved for human use, but for other indications. As described in detail below, these strategies are based on (1) the accumulating evidence that long COVID consistently impairs the functions of the PFC, (2) that COVID-19 infection increases kynurenine and cAMP-calcium signaling in brain, both of which may be especially damaging to PFC function given the unique molecular needs of PFC circuits, and (3) previous studies showing that N-acetyl-cysteine (NAC) and the $\alpha$2A-adrenoceptor ($\alpha$2A-AR) agonist, guanfacine, can reduce the effects of kynurenine and excessive cAMP-calcium signaling, respectively, and have been shown to protect the PFC from non-COVID

stressors in both animal models and human subjects. Thus, these agents may be helpful in treating the cognitive symptoms of long COVID.

## 2. Search Strategy and Selection Criteria

The references for this review were identified through searches of PubMed with the search terms "kynurenine" or "kynurenic acid", "long-COVID", "cognitive", "GCPII", "N-acetyl cysteine", "guanfacine", and "prefrontal cortex", with combined search terms of "cognitive", "long-COVID", and "meta-analysis" to help create Table 1. Articles were also identified through searches of the authors' own files. Only papers published in English were reviewed. The final reference list was generated on the basis of originality and relevance to the scope of this review, with an emphasis on review articles for summaries of PFC functions.

## 3. Long COVID Preferentially Afflicts the Cognitive Functioning of the Prefrontal Cortex

The PFC is a recently evolved part of the brain, situated rostral to the motor cortices in the frontal lobe, which expands enormously in primates [1–5]. The dorsolateral PFC (dlPFC) has been the focus of intensive research in monkeys and humans, as this area has the remarkable ability to represent information in the absence of sensory stimulation, the neural foundation for abstract thought and cognitive control. As summarized in Figure 1, this foundational mental capacity underlies a wide variety of functions, including abstract reasoning, working memory, and the executive functions, which include top-down control of attention, action, and emotion [1,6–16]. These abilities allow us to focus, divide or sustain our attention, e.g., as needed to multi-task, to inhibit inappropriate actions and impulses, and to plan and organize for the near and farther future. PFC circuits are needed for high-order, flexible decision-making, and the most rostral areas of the PFC mediate meta-cognitive functions such as insight and judgment about ourselves and others [1,17]. The rostal and lateral PFC regions project to the ventral and medial PFC to regulate emotion and motivation, allowing us to persist through difficulties, to regulate angry or aggressive impulses, and to have optimism about our future [18]. Lesions to the PFC impair these functions in humans and monkeys, including recent data showing that symptoms of PTSD and depression correlate with synapse loss in the dlPFC [19].

Many studies are finding that long COVID especially afflicts the functions of the PFC, in what patients often describe as "brain fog". There are consistent impairments in working memory, abstract reasoning, and the executive abilities, interfering with top-down control of attention (e.g., impaired concentration and multi-tasking), as well as regulation of actions and emotional state [20–25]. As described in Table 1, meta-analyses and reviews of this rapidly emerging field have emphasized the consistent impairment in executive functions [23,26–31], with the relative sparing of memory recognition, which, in the context of impaired encoding and recall, is consistent with impaired PFC function [21]. Becker and colleagues describe this constellation of deficits as having "considerable implications for occupational, psychological, and functional outcomes." [21]. The functional impairments in executive functioning are consistent with brain imaging studies, which show frontotemporal hypoperfusion in patients hospitalized with COVID-19 [20]. Long COVID is also commonly associated with symptoms of depression and anxiety [23], which may involve loss of PFC top-down regulation of emotion [19,32]. While there is some controversy regarding the relationships of severity of illness, age, or sex with the degree or frequency of cognitive impairments, e.g., deficits greater in young [33] vs. aged [34], there is general agreement that impairments in recall memory, working memory, attention regulation, and other executive functions are common (Table 1). It is clear that cognitive deficits can be problematic even after even mild illness [28,34,35] and that cognitive deficits may be related to inflammatory markers in plasma such as cytokines [36] and kynurenine [37]. There is also recent evidence that severe COVID infection can cause increased calcium dysregula-

tion, kynurenic acid expression, and tau hyperphosphorylation in brain, suggesting that it may increase risk of future Alzheimer's disease [38].

Why are the functions of the PFC so afflicted in long COVID, while other functions (e.g., visuospatial abilities, sensory perception, with the exception of olfaction) are so resilient? New data from animal studies indicate that the PFC is particularly vulnerable to the effects of stress and inflammation [39,40], which may help to explain why PFC deficits are such a common component of long COVID mental changes.

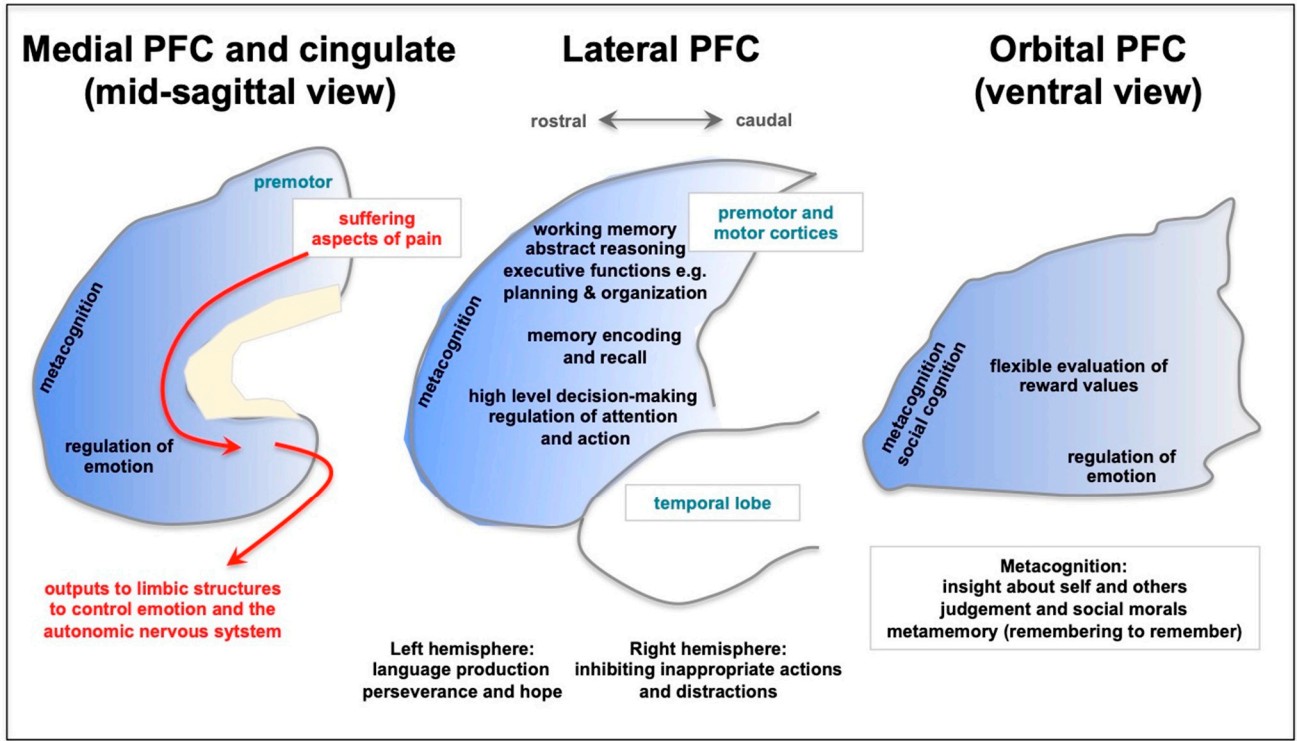

**Figure 1.** A schematic depiction of the functions of the prefrontal cortex (PFC), as illustrated on the medial, lateral, and ventral surfaces. There is a general topography, where representation of the external world is performed by lateral PFC of and the internal world by the ventral and medial PFC. The ventral (orbital) PFC flexibly represents our internal state, e.g., whether something is rewarding. The most caudal aspects of the medial surface also contain the cingulate cortices, which can activate emotional responses and are a key aspect of the suffering aspects of pain that are overactive in depression. Conversely, metacognitive functions such as insight about ourselves and others are localized most rostrally in the frontal pole. The lateral PFC generates many higher cognitive operations, including working memory, abstract thought, the executive functions, regulation of attention and action, and flexible decision-making. The PFC also assists with memory encoding and recall. The rostral and lateral PFC provides thoughtful "top-down" control of emotion through connections into the medial PFC. As described in the text, many of these PFC functions are preferentially afflicted in long COVID.

**Table 1.** Examples of recent reviews that describe PFC deficits associated with long COVID.

| Citation | Summary |
|---|---|
| Vanderlind et al., 2021 [23] | "Thirty-three studies met inclusion/exclusion criteria for review. Emerging findings link COVID-19 to cognitive deficits, particularly attention, executive function, and memory. Psychiatric symptoms occur at high rates in COVID-19 survivors, including anxiety, depression, fatigue, sleep disruption, and to a lesser extent posttraumatic stress." |
| Perrottelli et al., 2022 [26] | "The available evidence revealed the presence of impairment in executive functions, speed of processing, attention and memory in subjects recovered from COVID-19." |
| Zawilska and Kuczyńska, 2022 [27] | "Fatigue and cognitive dysfunction, such as concentration problems, short-term memory deficits, general memory loss, a specific decline in attention, language and praxis abilities, encoding and verbal fluency, impairment of executive functions, and psychomotor coordination, are amongst the most common and debilitating features of neuropsychatric symptoms of post COVID syndrome." |
| Newhouse et al., 2022 [41] | "Many of these symptoms [42] are neuropsychiatric, such as inattention, impaired memory, and executive dysfunction; these are often colloquially termed "brain fog"." |
| Bertuccelli et al., 2022 [28] | "Memory, attention, and executive functions appeared to be the most affected domains. Delayed recall and learning were the most impaired domains of memory. Among the executive functions, abstraction, inhibition, set shifting, and sustained and selective attention were most commonly impaired." |
| Houben and Bonnechère, 2022 [29] | Meta-analysis showed impairments in attention, executive functioning, and verbal memory (recall) |
| Zeng et al., 2023 [30] | "Individuals with severe infection suffered more from PTSD, sleep disturbance, cognitive deficits, concentration impairment, and gustatory dysfunction. Survivors with mild infection had high burden of anxiety and memory impairment after recovery." |
| Ceban et al., 2022 [31] | "The proportion of individuals exhibiting cognitive impairment was 0.22 (95% CI, 0.17, 0.28; $p < 0.001$; $n = 13,232$; $I^2 = 98.0$). Moreover, narrative synthesis revealed elevations in proinflammatory markers and considerable functional impairment in a subset of individuals." |

Many studies are finding that long COVID especially afflicts the functions of the PFC, in what patients often describe as "brain fog". There are consistent impairments in working memory, abstract reasoning, and the executive abilities, interfering with top-down control of attention (e.g., impaired concentration and multi-tasking), as well as regulation of actions and emotional state [20–25]. Meta-analyses have emphasized the consistent impairment in executive functions [23], with the relative sparing of memory recognition, which in the context of impaired encoding and recall is consistent with impaired PFC function [21]. Becker and colleagues describe this constellation of deficits as having "considerable implications for occupational, psychological, and functional outcomes." [21]. The functional impairments in executive functioning are consistent with brain imaging studies, which show frontotemporal hypoperfusion in patients hospitalized with COVID-19 [20]. Long COVID is also commonly associated with symptoms of depression and anxiety [23],

which may involve loss of PFC top-down regulation of emotion [19]. Emerging data from animal studies indicate that the PFC is particularly vulnerable to the effects of stress and inflammation, which may help to explain why PFC deficits are such a common component of long COVID mental changes.

## 4. The Prefrontal Cortex Is Especially Vulnerable to Physiological and Psychological Stressors

PFC dysfunction occurs with multiple physiological and psychological stressors. Deficits in working memory and executive dysfunction are a major component of the sustained cognitive deficits following traumatic brain injury (TBI) in both human patients [43–46] and rodents [47,48]. The studies in rodents showed that there are multiple chemical and structural changes in the rat PFC, even though the concussive site was to the posterior cortex and not the PFC itself [47,48]. Similar patterns were seen with sustained hypoxia, which impairs working memory and the executive functions in both humans [49] and rodents [50], the latter studies showing pronounced chemical and structural changes in the PFC [50,51].

PFC cognitive deficits are also produced by psychological stressors, particularly when the subject feels that they have no control over the stressor [52]. Research in both humans and animals has shown that exposure to even an acute, uncontrollable stress rapidly impairs PFC working memory abilities and reduces the activity of the dlPFC [53]. With chronic exposure to uncontrollable psychological stress, there are additional architectural changes, with loss of dendrites and spines (and thus neural connections) from PFC neurons which correlate with impaired working memory and attentional control [39,54,55]. Brain imaging of human subjects also shows weaker dlPFC connectivity with chronic stress exposure [54] as well as loss of PFC gray matter [56]. Exposure to a traumatic psychological stressor can induce symptoms of PTSD, which are related to reduced dlPFC synapses [19].

As reviewed below, data from animal studies suggest that psychological and physiological stressors activate similar molecular signaling pathways that may be especially deleterious in the PFC due to its unusual molecular dependencies.

## 5. Unique Neurotransmission and Neuromodulation Renders PFC Circuits Especially Vulnerable to Stress and Inflammation

dlPFC circuits have special circuits and molecular properties that allow them to generate and sustain neuronal firing without any sensory stimulation. The circuits have extensive, local, recurrent excitation, whereby the neurons excite each other in order to keep firing without needing external stimulation [6]. This recurrent excitation relies on unusual neurotransmission and neuromodulation, summarized in Figure 2, with the molecular machinery to magnify calcium [53]. Although the calcium is necessary to maintain neuronal firing, if the levels are too high, they open nearby $K^+$ channels to functionally disconnect circuits, which markedly reduces neuronal firing and impairs cognition [53]. With sustained high levels of calcium, inflammatory mechanisms are evoked that lead to loss of spines and dendrites. These properties render PFC neurons very vulnerable to atrophy under conditions of inflammation [40], where molecular changes induced by COVID-19 would be especially deleterious to the dlPFC.

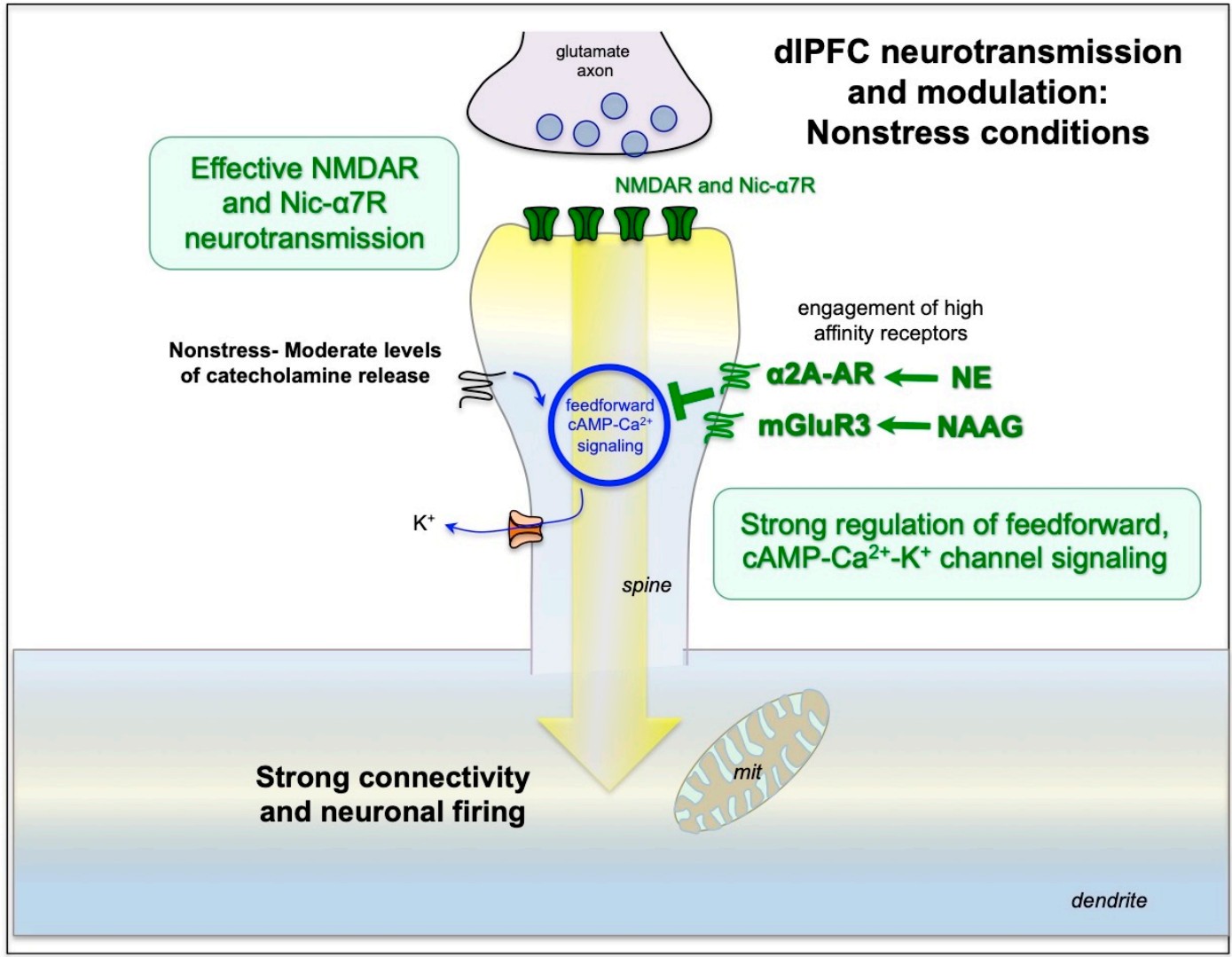

**Figure 2.** The recently evolved neuronal circuits in the primate dlPFC have unusual neurotransmission and neuromodulation for healthy cognitive functioning. In contrast to typical glutamate circuits which rely heavily on AMPAR stimulation, dlPFC neurotransmission relies on NMDAR and nic-α7R with little contribution from AMPAR [57,58]. NMDAR opening can only occur in a depolarized synaptic membrane, which may be sustained by both nic-α7R stimulation and by higher calcium levels near the synapse. Layer III dlPFC spines express the molecular machinery to magnify calcium signaling near the synapse, where cAMP-PKA signaling is positioned to increase internal calcium release from the SER (smooth endoplasmic reticulum) through ryanodine receptors (RyR) and IP3 receptors (IP3R) [53]. Calcium in turn increases cAMP production, thus producing feedforward signaling that must be tightly regulated by α2A-AR and mGluR3, as described below. (The feedforward signaling is represented by the blue circle.) Moderate levels of cAMP-calcium signaling are necessary for dlPFC neuronal firing during working memory, but higher levels can reduce firing by opening nearby potassium ($K^+$) channels. Under nonstress conditions, there are moderate levels of norepinephrine (NE) and dopamine release. Moderate levels of NE engage high-affinity α2A-AR (rather than lower-affinity α1-AR or β-AR), which are localized on dendritic spines [59]. mGluR3 are also on spines and are stimulated by both glutamate and NAAG (N-acetylaspartylglutamate), which is co-released with glutamate and is selective for mGluR3 [60]. α2A-AR and mGluR3 signaling regulate feedforward cAMP-calcium signaling, reducing cAMP-calcium opening of nearby $K^+$ channels [53,61]. This strengthens network connectivity and increases the neuronal firing needed for higher cognition. mit = mitochondrion.

Unusual neurotransmission—The persistent firing of dlPFC neurons during working memory differs from typical glutamate neurotransmission in its great dependence on NMDA receptors (NMDAR) rather than AMPA receptors (AMPAR). Classic glutamate neurotransmission depends heavily on AMPAR, where glutamate stimulation of AMPAR fluxes $Na^+$ ions to depolarize the synaptic membrane. This depolarization ejects $Mg^{2+}$ from the NMDAR pore, permitting NMDAR neurotransmission [53]. As NMDARs flux both $Na^+$ and $Ca^{2+}$, they can be particularly important for neuroplasticity. AMPAR neurotransmission has a rapid on- and offset, and thus is especially appropriate for encoding sensory events [58].

In contrast to traditional glutamate synapses, the recurrent excitation in dlPFC circuits has little reliance on AMPAR, but it relies heavily on NMDAR neurotransmission, including NMDAR with GluN2B subunits, which close slowly and flux high levels of calcium [57]. The permissive role normally played by AMPAR is instead played by acetylcholine, including stimulation of nicotinic $\alpha$7 receptors (nic-$\alpha$7R), which reside within the glutamate synapse in dlPFC circuits [58]. The reliance on acetylcholine makes dlPFC especially dependent on arousal state as acetylcholine is released during waking and not during deep sleep [58]. As described below, this dependence on NMDAR and nic-$\alpha$7R makes dlPFC circuits particularly vulnerable to blockade by kynurenic acid inflammatory signaling.

Unusual neuromodulation—In most neurons, increased cAMP-calcium signaling strengthens neuronal function and synaptic connections, e.g., as occurs in hippocampus for long-term increases in synaptic strength [53]. However, in layer III dlPFC, there is an enrichment of potassium ($K^+$) channels on spines that are opened by cAMP-PKA-calcium signaling, which weakens rather than strengthens dlPFC connectivity and function [53]. As stress and inflammation increase cAMP-calcium signaling, the concentration of $K^+$ channels opened by cAMP-calcium signaling on layer III dlPFC spines makes these neurons particularly vulnerable to stress and inflammation [53].

As described above, dlPFC dendritic spines contain the cAMP-PKA molecular machinery to magnify calcium signaling near the PSD to support persistent neuronal firing [53]. As shown in Figure 2, there is evidence of extensive feedforward, cAMP-PKA-calcium signaling, whereby cAMP-PKA increases calcium release from the smooth endoplasmic reticulum (SER), and increased calcium drives more cAMP signaling. Moderate levels of cAMP-calcium signaling are needed to support sustained neuronal firing needed for working memory and top-down control. However, high levels of cAMP-PKA-calcium signaling reduce firing by opening nearby $K^+$ channels (e.g., HCN-Slack, KCNQ2) on spines (Figure 2). Low levels provide negative feedback to prevent seizures, but high levels of $K^+$ channel opening, as occurs with stress exposure, markedly reduce neuronal firing and impair working memory and top-down control [53]. This occurs with stress exposure when there are high levels of catecholamine release in PFC [62,63], which engage low-affinity receptors (e.g., $\alpha$1-AR, Figure 3) that increase cAMP-calcium-$K^+$ signaling and impair working memory [53]. Sustained elevations in cAMP-PKA-calcium-$K^+$ signaling, i.e., continued weakening of synaptic connections, lead to removal of spines and dendrites, involving calcium-induced inflammatory mechanisms (Figure 3) [39].

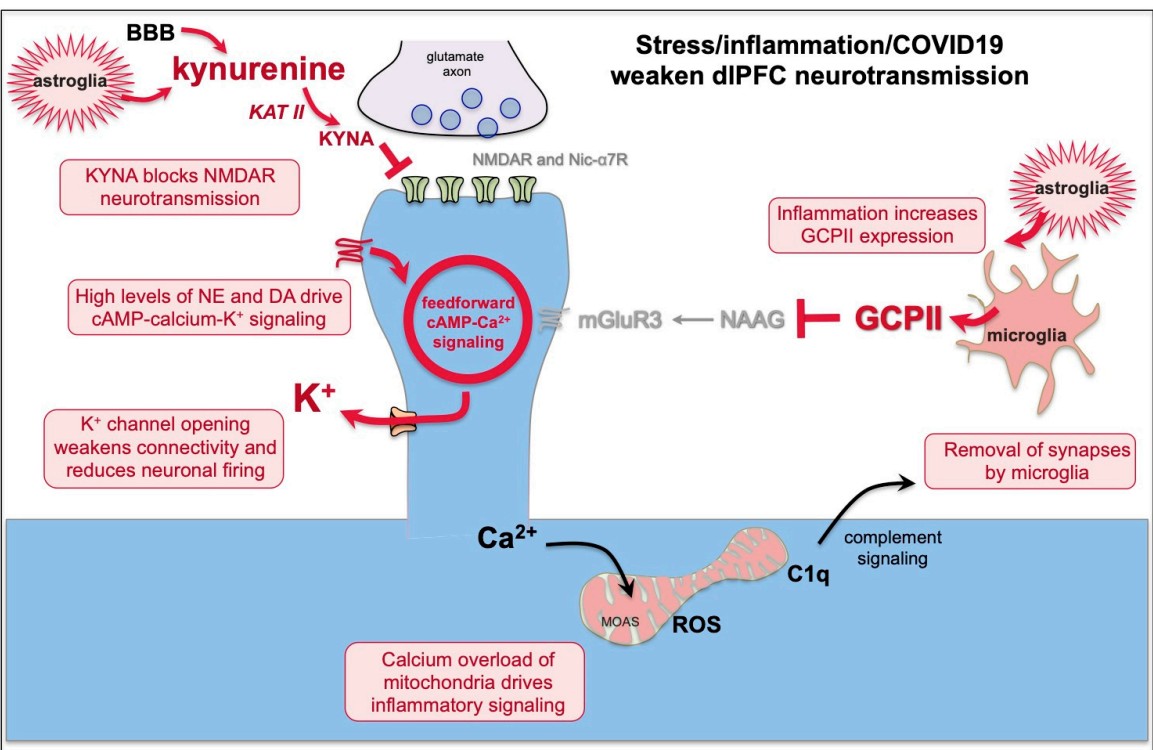

**Figure 3.** The effects of stress and inflammation on dlPFC networks. Uncontrollable stress and/or inflammation weaken dlPFC network connections in multiple ways. (1) Kynurenine is metabolized from tryptophan under inflammatory conditions in astrocytes and is also actively taken up from the blood through the blood brain barrier (BBB). In the brain, kynurenine can be further metabolized by KAT II to KYNA, which blocks both NMDA and nic-$\alpha$7R, the two receptors needed for dlPFC neurotransmission. Thus, KYNA impairs PFC function. (2) Both psychological and physiological stressors cause high levels of catecholamine release in the PFC, which engage low-affinity $\alpha$1-AR or $\beta$-AR as well as high levels of D1R stimulation, driving cAMP-calcium opening of large numbers of K$^+$ channels, which weakens connectivity and reduces neuronal firing. (3) Microglial and astrocytic production of GCPII with inflammation catabolizes NAAG, reducing mGluR3 regulation of cAMP-calcium signaling, thus decreasing neuronal firing. (4) Sustained high levels of cytosolic calcium and calcium release from the SER with chronic stress/inflammation also cause calcium overload of mitochondria, leading to complement C1q signaling to microglia to remove the spine. COVID-19 infection is known to increase both KYNA and GCPII in brain and to increase cAMP-PKA-calcium signaling (see text). MOAS = mitochondria on a string, an abnormal mitochondrial phenotype consistent with calcium overload and an inflammatory state. ROS = reactive oxygen species.

Under healthy conditions, cAMP-PKA drive on calcium-K$^+$ channel signaling is regulated by $\alpha$2A-adrenoceptors ($\alpha$2A-AR) [59] and type 3 metabotropic glutamate receptors (mGluR3s) [61] on spines, which inhibit cAMP production, strengthen connectivity and enhance neuronal firing (Figure 2). mGluR3 signaling expands in primates and is particularly important for human cognition [64]. mGluR3 are stimulated by both glutamate and NAAG (N-acetylaspartylglutamate), which is co-released with glutamate and is selective for mGluR3. As summarized in Figure 3, NAAG is catabolized by GCPII (glutamate carboxypeptidase II), which is increased under conditions of inflammation, reducing mGluR3 regulation of cAMP-calcium-K$^+$ channel signaling, which decreases dlPFC neuronal firing and impairs working memory [61,65].

## 6. Stress and Inflammatory Signaling in Brain—Similar Pathways Activated by Psychological and Physiological Stressors, Including Activation by COVID-19

Emerging data suggest that stress and inflammation activate many of the same intracellular signaling pathways, and that they may have interactive effects, e.g., whereby inflammation reduces regulation of the stress response, which in turn causes more inflammation [53]. These vicious cycles may trap the brain in a detrimental state, requiring treatment to restore normal function.

As described above, rodent studies of psychological stressors, such as restraint stress or conditioned fear, increase catecholamine release in the PFC, which activates feedforward cAMP-calcium signaling to open $K^+$ channels, functionally disconnecting dlPFC circuits and rapidly impairing PFC cognitive abilities (Figure 3). With chronic psychological stress in rodents, sustained elevations in cAMP-PKA-calcium-$K^+$ signaling leads to removal of spines and dendrites, which correlates with impaired PFC function [55,66]. Similar findings can be seen in humans, where chronic psychological stress is associated with reductions in PFC gray matter [56] and reduced dlPFC functional connectivity [54]. The loss of spines likely involves calcium overload of mitochondria, initiating complement inflammatory signaling to microglia to remove the weakened spine (Figure 3) [39].

Although psychological and physiological stressors are remarkably different (e.g., fear of losing your job vs. an hypoxic event), they appear to activate many of the same molecular signaling pathways in PFC and these are worsened/mimicked by inflammation, which dysregulate stress signaling pathways. Data from animal and human studies show that both psychological and physiological stressors increase catecholamine release and/or induce high levels of cAMP-PKA-calcium signaling in PFC. For example, TBI to the posterior cortex, such as psychological stress, increases catecholamine release, $\alpha$1-AR expression, and PKA-calcium signaling in the rodent PFC and causes working memory deficits, even though the injury occurred distant to the PFC [47,67–69]. Similarly, hypoxia increases catecholamine release and calcium dysregulation in the rodent PFC and impairs working memory [50,51]. Both conditions, similarly to chronic psychological stress [55,66], cause loss of dendrites and spines in PFC, eroding PFC connectivity [48,51].

A recent study of brains of patients who died from COVID-19 found similar changes to those reported with psychological and other physiological stressors, with marked increases in cAMP-PKA signaling and calcium dysregulation [38]. This study also showed elevations in GCPII and in kynurenine-kynurenic acid signaling [38], which may particularly impair PFC function. As patients who died from COVID-19 also had significant hypoxia, it is important to note that hypoxia on its own induces calcium dysregulation and spine loss in the rat PFC, as described above. Thus, we cannot know if the brain changes in patients who died from COVID-19 were caused by the viral infection, hypoxia, or both. The finding of sustained PFC cognitive deficits in patients with long COVID who did not have hypoxic episodes suggests that at least some of the brain changes are due to an inflammatory response to the virus, independent of hypoxia, e.g., due to increases in systemic kynurenine production.

## 7. Kynurenic Acid Signaling Is Increased by Inflammation, Including by COVID-19, and Can Reduce NMDAR and nic-$\alpha$7R Neurotransmission

Kynurenine is metabolized from tryptophan under conditions of inflammation (Figures 3 and 4). Kynurenine is synthesized locally in brain by astrocytes and is also actively taken up from the blood into the brain, where it can be further metabolized by KAT II (kynurenine aminotransferase II) into kynurenic acid (KYNA), or by KMO (kynurenine 3-monooxygenase) to produce quinolinic acid [70]. KYNA blocks both NMDAR and nic-$\alpha$7R, while quinolinic acid stimulates NMDA R [70]. Researchers examining the mechanisms underlying excitotoxicity have shown that under conditions of high glutamate release, e.g., during a stroke or traumatic brain injury, or when the brain is sliced for in vitro recordings, quinolinic acid stimulation of NMDAR contributes to toxic actions and

neuronal death, while KYNA blockade of NMDAR is protective, preventing cell death [71]. Thus, KYNA is often included in the bath during in vitro slice recordings.

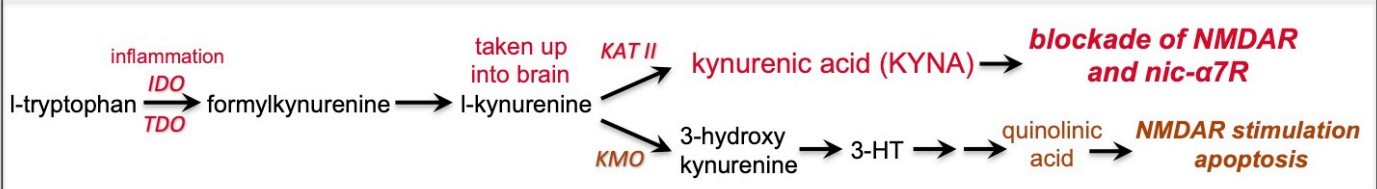

**Figure 4.** A summary of kynurenine signaling, which is increased by COVID-19 infection. Kynurenine is metabolized from tryptophan by the immune system and by astrocytes under inflammatory conditions. It is actively taken up into the brain, where it can be further metabolized to KYNA, which blocks NMDAR and nic-α7R or to quinolinic acid, which stimulates NMDAR.

However, researchers have also examined the effects of KYNA under non-excitotoxic conditions, e.g., in chronic neuroinflammatory disorders when there is normal or even reduced glutamate release but elevated kynurenine signaling. Under these conditions, KYNA blockade of NMDAR is detrimental as it impairs cognition [72]. For example, schizophrenia is associated with lower or normal levels of glutamate in the PFC [73] and with increased neuroinflammation [74], including elevated KYNA expression in the dlPFC that correlates with impaired dlPFC function [75]. As dlPFC neurotransmission depends heavily on NMDAR and nic-α7R [57,58], KYNA blockade of both of these receptors would be particularly detrimental to dlPFC function.

It is likely that increased KYNA expression in the PFC contributes to the cognitive deficits of long COVID-19. There are extensive data showing that COVID19 infection increases kynurenine levels in plasma that correlate with cytokine levels and are particularly high in men [42,76–79]. Similar findings have been seen in macaques infected with SARS-CoV-2 [80]. Post-mortem evaluations of brains of patients who died from COVID infection show very large elevations in KYNA expression [38], demonstrating its kynurenine conversion to KYNA in brain. In patients with long COVID, plasma kynurenine levels remain high [81]. Most pertinent to the current review, cognitive deficits in long COVID highly correlate with plasma kynurenine levels [37], as do symptoms of anxiety and stress, which are also signs of PFC dysfunction [82].

Studies in animals support the hypothesis that elevated KYNA in the PFC impairs cognitive functioning. Treatments that increase KYNA in the PFC impair cognitive function in rats [83,84]. Conversely, treatments that reduce KYNA expression in brain improve cognitive performance in monkeys, rats, and mice [85,86]. Our preliminary data show that iontophoresis of KYNA markedly reduces dlPFC neuronal firing in monkeys performing a working memory task, consistent with this hypothesis (M. Wang, unpublished). Thus, agents that reduce KYNA production may be especially helpful in restoring dlPFC cognitive functioning in patients with long COVID.

## 8. GCPII Expression Is Increased by Inflammation, Including by COVID-19, and Reduces mGluR3 Regulation of cAMP-Calcium Signaling

As summarized in Figure 3, the large production of GCPII under inflammatory conditions destroys NAAG and reduces mGluR3 signaling. Elevated GCPII has been seen in multiple inflammatory conditions [87–89], including elevated GCPII in rodent brain following hypoxia from stroke [90]. Elevated GCPII is associated with impaired cognition in animal models, where GCPII inhibition can restore cognitive abilities [91,92]. Most pertinent to the current review, there are very large increases in GCPII expression in the brains of patients who died from COVID-19 [38]. Elevations in GCPII would be particularly harmful to the proper functioning of the primate dlPFC, as mGluR3 signaling is needed to regulate feedforward cAMP-calcium-K$^+$ signaling in dlPFC spines (Figure 3). Thus, elevated GCPII

signaling may have contributed to the large increases in cAMP-PKA-calcium signaling in the brains of patients who died from COVID-19 [38].

In summary, the data suggest that KYNA blockade of NMDAR and nic-$\alpha$7R neurotransmission, as well as GCPII dysregulated cAMP-calcium signaling, may especially weaken dlPFC circuit firing by (1) reducing neurotransmission and (2) increasing cAMP-PKA-calcium opening of K$^+$ channels, functionally disconnecting dlPFC circuits (Figure 3). Sustained elevations in feedforward calcium signaling would also lead to loss of spines and dendrites through calcium overload of nearby mitochondria and the initiation of inflammatory signals to microglia to engulf the afflicted spine (Figure 3). Thus, treatments that inhibit the production of KYNA and/or that restore regulation of cAMP-calcium signaling in dlPFC may be effective in treating the PFC cognitive symptoms of long COVID.

## 9. Strategies for Treatment Based on the Neuroscience

Specific inhibitors of kynurenine [85] or GCPII [93] signaling are currently under development, but are not yet available for human use. The following describes two compounds that are FDA-approved for other indications that may be useful in normalizing dlPFC physiology and restoring cognitive function by reducing the production of KYNA and increasing the regulation of cAMP-calcium signaling in dlPFC.

N-acetylcysteine (NAC)—NAC is FDA-approved for the treatment of acetaminophen overdose [94], which has antioxidant properties, e.g., by increasing endogenous glutathione levels [94] and protecting mitochondria from calcium overload [95]. Animal models of TBI have shown that NAC reduces neuropathology [95,96], and NAC is now in widespread, off-label use to treat TBI in patients, with few side effects [97,98]. NAC has also been proposed as an adjunct treatment for acute COVID-19 infection based on its antioxidant properties [99]. However, it has recently been discovered that NAC also inhibits KAT-II, the enzyme that converts kynurenine to kynurenic acid. NAC treatment significantly reduced KYNA levels in the rodent PFC [100] and chronic NAC treatment reduced KYNA and protected cognitive function in mice treated with kynurenine [86]. Thus, its ability to inhibit KAT II, in addition to its antioxidant properties, may be helpful in restoring NMDAR and nic-$\alpha$7R neurotransmission in the dlPFC of patients with PFC cognitive deficits due to long COVID (Figure 5).

Guanfacine—Guanfacine is a selective $\alpha$2A-AR agonist that was originally FDA-approved for the treatment of hypertension in 1986 and more recently FDA-approved for the treatment of Attention Deficit Hyperactivity Disorder (ADHD) in 2009 [101]. ADHD is characterized by weak regulation of attention and impulse control, involving impaired development and/or functioning of the PFC [102]. The extended release formulation of guanfacine (Intuniv) is often used to treat ADHD in children [101] and younger adults [103], given the faster drug metabolism in younger patients. As summarized in Figure 5, guanfacine's beneficial actions likely involve both strengthening of dlPFC neuronal network connections and reducing neuroinflammation. Guanfacine's ability to strengthen dlPFC function has been studied at the cellular level, where $\alpha$2A-ARs are localized post-synaptically on dlPFC spines near the HCN channels that weaken network connectivity [59]. Local or systemic administration of guanfacine enhances dlPFC neuronal firing and cognitive functions by inhibiting cAMP-calcium-K$^+$ channel signaling, and thus strengthening network connectivity and PFC activity [104]. Guanfacine and other $\alpha$2A-AR agonists are also anti-inflammatory, deactivating both microglia [105,106] and macrophages [107]. Guanfacine has been shown to enhance and/or protect PFC structure and function in a variety of animal models, including rescuing PFC cognitive functioning and spine density from the deleterious effects of chronic psychological stress [55] or hypoxia [50,51]. Guanfacine has also been used off-label in human subjects to treat cognitive deficits caused by TBI [44,108], hypoxia [109], and psychological stress [110], and it has established its safety over decades of use. It is currently being tested for the treatment of delirium (https://clinicaltrials.gov: NCT04742673 and NCT04578886), a condition that would benefit from both anti-inflammatory and PFC-strengthening actions.

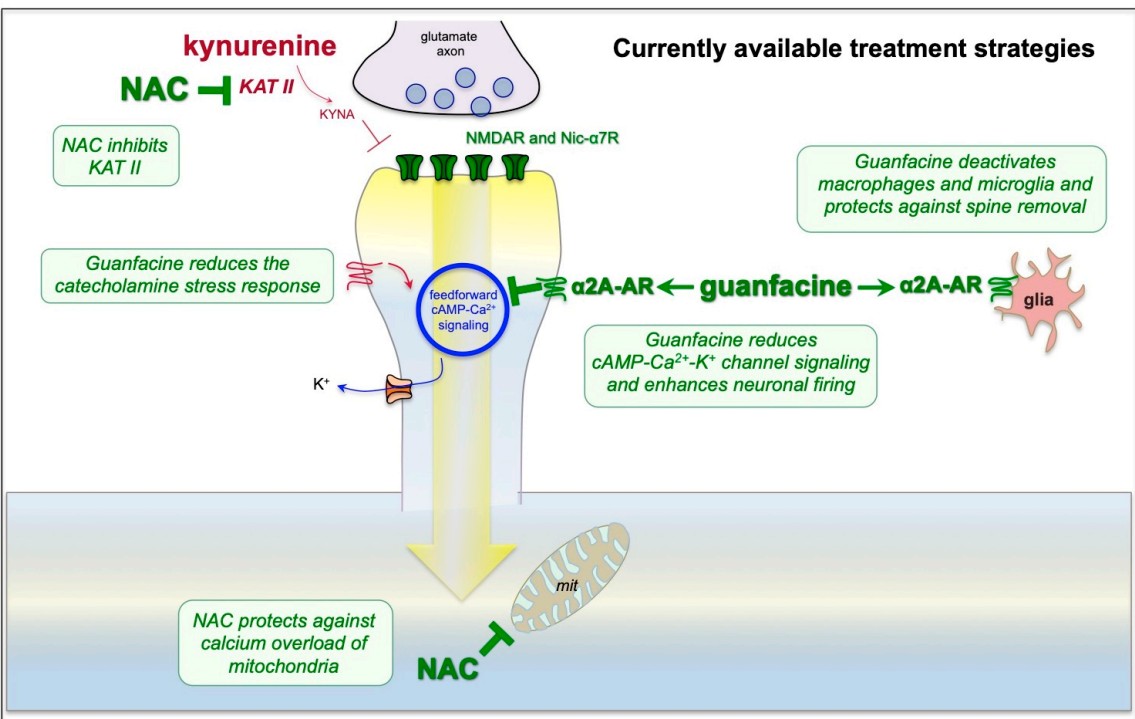

**Figure 5.** Potential treatments for the cognitive deficits of long COVID can normalize dlPFC physiology and thus may be helpful in treating the cognitive symptoms of long COVID. NAC may normalize dlPFC neurotransmission by blocking the production of KYNA, while the α2A-AR agonist, guanfacine, may restore regulation of cAMP-calcium-K$^+$ signaling by stimulating α2A-AR on spines. NAC is also known to have antioxidant properties that would protect mitochondria, and α2A-AR agonists deactivate microglia and macrophages and thus have general anti-inflammatory properties.

Given the established safety of both NAC and guanfacine and the urgent need for treatment of cognitive impairment in long COVID, these agents may provide interim relief while waiting for FDA-approved therapies. Our preliminary, open-label findings [111] indicate improvement in cognitive functioning in patients with long COVID following treatment with a combined daily treatment with both NAC (600 mg) and guanfacine (1 mg, escalating to 2–3 mg as needed). Placebo-controlled trials are needed to establish efficacy, but the scientific rationale for their use, outlined here, suggests they may have immediate utility in patients with urgent need.

## 10. Outstanding Questions and Summary

The current pandemic has created a sudden and urgent need for treatments that may alleviate the sustained cognitive deficits caused by COVID-19 infection. Understanding how COVID-19 affects the brain is an arena where much still needs to be learned, especially with regard to those with less severe infection who nonetheless retain debilitating cognitive deficits. This is particularly challenging in living individuals, where in vivo brain imaging of inflammation is very limited, and analyses often rely on distal measures such as CSF or plasma measures of inflammation.

There is also an outstanding need for new treatments aimed at this indication, such as GCPII inhibitors approved for human use, which will likely require many years for successful development. There is also a need for placebo-controlled trials of existing compounds approved for other indications, such as NAC and guanfacine, as described in this review. However, it takes many years to acquire funding and to conduct sufficiently large trials to have confident results. As the cognitive deficits of long COVID are often sufficiently severe to impede work and home life, rational strategies for treatment are needed to help those whose lives have been upended by the ongoing pandemic.

In summary, the cognitive deficits of long COVID particularly involve deficits in dlPFC function. The unusual neurotransmission and neuromodulation of dlPFC circuits make them especially vulnerable to inflammation, including KYNA blockade of NMDAR and nic-α7R, the receptors needed for dlPFC neurotransmission, and the inflammatory factors such as GCPII that lessen regulation of magnified calcium signaling, thus increasing calcium's toxic actions. Compounds such as NAC and guanfacine, which block the production of KYNA and increase the regulation of cAMP-calcium signaling, respectively, may be helpful in restoring dlPFC physiology and function in the absence of FDA-approved treatments for this indication.

**Author Contributions:** A.F.Z., A.F.T.A. and M.W. wrote/revised this review based on their expertise in the clinical and basic neuroscience, respectively. All authors have read and agreed to the published version of the manuscript.

**Funding:** A.F.Z. is supported by 1P30AG066508-01. A.F.T.A. and M.W. are supported by NIH AG061190 and Neuronex NSF 2015276.

**Institutional Review Board Statement:** Not applicable.

**Informed Consent Statement:** Not applicable.

**Data Availability Statement:** Not applicable.

**Acknowledgments:** The authors acknowledge that there were no additional external funding for the work presented.

**Conflicts of Interest:** The authors declare no conflict of interest.

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
