# Peer review of "Scientific Rationale for the Treatment of Cognitive Deficits from Long COVID"

_2035-8377, doi:10.3390/neurolint15020045_

Round 1

Reviewer 1 Report

The manuscript reports an interesting review of the literature about the possible strategies that might be evaluated and implemented for the treatment of long-COVID. The review covers areas that are interesting for clinicians and researchers, showing possible future strategies. The main goal of the manuscript is to evaluate possible therapeutic strategies based on the existing literature on cognitive impairment that might be applied in patients with "brain fog" after COVID-19 infection. I think the authors have done a great job because the paper might be an interesting starting point for future studies. I do not have specific concerns. The conclusions are coherent. The quotes are adequated. I have just some advice for the manuscript:

- Please include more information about what is known about brain fog after COVID-19. This might be the rationale for your paper.

- It is not clear to me - based on Figure 1 - if the authors have considered the entire brain or just the cortex

.- I know that the manuscript is not a systematic review, but the authors might consider reporting the number of papers revised and the rate included in order to give an idea of the existing literature

- Please state clearly the limit linked to the nature of the methodology that might miss some useful paper. 

- Please consider including a table with the summarized results to make the paper more useful for clinicians.

Author Response

“The manuscript reports an interesting review of the literature about the possible strategies that might be evaluated and implemented for the treatment of long-COVID. The review covers areas that are interesting for clinicians and researchers, showing possible future strategies. The main goal of the manuscript is to evaluate possible therapeutic strategies based on the existing literature on cognitive impairment that might be applied in patients with "brain fog" after COVID-19 infection. I think the authors have done a great job because the paper might be an interesting starting point for future studies. I do not have specific concerns. The conclusions are coherent. The quotes are adequate. I have just some advice for the manuscript:”

Response: We are very grateful to Reviewer 1 for appreciating our manuscript, and for such helpful comments which have improved the paper.

“- Please include more information about what is known about brain fog after COVID-19. This might be the rationale for your paper.”

Response: We have expanded the paragraph describing the cognitive symptoms of long-COVID, and have also included descriptions in Table 1.

  • “It is not clear to me - based on Figure 1 - if the authors have considered the entire brain or just the cortex”

Response: The paper focuses on the prefrontal cortex, as these are the functions most consistently altered by long-COVID. The revised figure hopefully makes this more clear.

“- I know that the manuscript is not a systematic review, but the authors might consider reporting the number of papers revised and the rate included in order to give an idea of the existing literature”

Response: Research on the cognitive deficits of long-COVID is a rapidly expanding field, with an exponential rise in the number of papers addressing this topic (over 350 at last count). There are already multiple reviews and meta-analyses aimed at capturing these emerging data, which we have now summarized in the new Table 1. A thorough review of this field is thus beyond the scope of the current manuscript, which instead aims to describe the neurobiological mechanisms by which inflammatory responses may particularly target prefrontal cortical circuits, including strategies for treatment. Please note that all of the reviews of research in Table 1 cite consistent deficits in executive functions, memory recall and/or attention regulation, i.e. the functions of the prefrontal cortex.

“- Please state clearly the limit linked to the nature of the methodology that might miss some useful paper. “

Response: The section entitled “2. Search strategy and selection criteria” has been expanded, but as noted above, it is not our intention to provide a comprehensive review of cognitive deficits in long-COVID, as these reviews are already available (new Table 1). As we have been researching the molecular regulation of prefrontal cortical circuits for decades, we hope that we have included the most relevant papers for this audience.

“- Please consider including a table with the summarized results to make the paper more useful for clinicians.”

Response: As Reviewer 2 also requested a Table, we have now been added a Table summarizing recent reviews of this field to help to emphasize that impairments in executive function, recall memory and attention regulation i.e. impairments in prefrontal abilities, are common symptoms of long-COVID cognitive deficits.

Reviewer 2 Report

My suggestions:

1. I would describe the long-covid symptoms in a little bit more in detail

2. I would draw Figure 1 a little differently. Instead of the entire brain, I would just highlight the prefrontal cortex. I would highlight all parts of the prefrontal cortex, and summarize the function region-wise. I would also highlight, which functions may be disturbed in case of long-covid. 

3. Is it possible that long-covid "brain fog" could be a risk for cognitive dysfunctions, such as Alzheimer's disease or frontotemporal dementia?

4. I would add a table, which summarizes the potential therapeutic strategies, which could be useful for the cognitive dysfunctions, associated with long-covid symptoms. 

Author Response

We thank Reviewer 2 for such constructive suggestions to improve the paper.

“My suggestions:

  1. I would describe the long-covid symptoms in a little bit more in detail”

Response: We have enlarged this section and have also added Table 1 to add further descriptions and summaries of prefrontal cognitive deficits in long-COVID. Please note that there are now hundreds of papers on this topic, including many reviews, so it is not our intention to repeat these efforts. The reviews/meta-analyses are now emphasized in Table 1 for the interested reader.

“2. I would draw Figure 1 a little differently. Instead of the entire brain, I would just highlight the prefrontal cortex. I would highlight all parts of the prefrontal cortex, and summarize the function region-wise. I would also highlight, which functions may be disturbed in case of long-covid.”

Response: We have taken the Reviewer’s suggestion and recreated Figure 1 to focus only on the prefrontal cortex and to show the topographic relationships to function on the lateral, medial and ventral surfaces. As there has been no explicit examination of all of these functions in long-COVID, we have chosen to not highlight those already found to be impaired in long-COVID, as it could portray false negatives as this field continues to evolve. As Dr. Arnsten (an expert on primate prefrontal cortex) made this new figure, as well as helping us with the new Table, we have added her as an author to acknowledge her contributions.

“3. Is it possible that long-covid "brain fog" could be a risk for cognitive dysfunctions, such as Alzheimer's disease or frontotemporal dementia?”

Response: Yes, the emerging data indicate that COVID infection increases Alzheimer’s-like pathology in brain, including hyperphosphorylation of tau (Reiken et al. Alzheimer's-like signaling in brains of COVID-19 patients. Alzheimers Dement. 18: 955-65, 2022). This finding is now highlighted as follows:

“There is also recent evidence that severe COVID infection can cause increased calcium dysregulation, kynurenic acid expression and tau hyperphosphorylation in brain, suggesting that it may increase risk of future Alzheimer’s disease(Reiken et al., 2022).”

“4. I would add a table, which summarizes the potential therapeutic strategies, which could be useful for the cognitive dysfunctions, associated with long-covid symptoms. “

Response: As Reviewer 1 also requested a Table, we have now been added a Table summarizing recent reviews of this field to help to emphasize that impairments in executive function, recall memory and attention regulation i.e. impairments in prefrontal abilities, are common symptoms of long-COVID cognitive deficits.

Round 2

Reviewer 2 Report

The manuscript is acceptable now.